# An Efficient RSS Localization for Underwater Wireless Sensor Networks

**DOI:** 10.3390/s19143105

**Published:** 2019-07-13

**Authors:** Thu L. N. Nguyen, Yoan Shin

**Affiliations:** School of Electronic Engineering, Soongsil University, Seoul 06978, Korea

**Keywords:** received signal strength, localization, underwater wireless sensor network, linear regression, relational distance refinement

## Abstract

Localization is a key-enabling technology for many applications in underwater wireless sensor networks. Traditional approaches for received signal strength (RSS)-based localization often require uniform distribution for anchor nodes and suffer from poor estimates according to unpredictable and uncontrollable noise conditions. In this paper, we establish an RSS-based localization scheme to determine the location of an unknown normal sensor from a certain measurement set of potential anchor nodes. First, we present a practical path loss model for wireless communication in underwater acoustic environments, where anchor nodes are deployed in a random circumstance. For a given area of interest, the RSS data collection is performed dynamically, where the measurement noises and the correlation among them are taken into account. For a pair of transmitter and receiver, we approximate the geometry distance between them according to a linear regression model. Thus, we can obtain a quick access for the range information, while keeping the error, the communication head and the response time low. We also present a method to correct noises in the distance estimate. Simulation results demonstrate that our localization scheme achieves a better performance for certain scenario settings. The successful localization probability can be up to 90%, where the anchor rate is fixed at 10%.

## 1. Introduction

Recently, the development of underwater wireless sensor networks (UWSNs) has openned great possibilities in several civilian and military applications such as event detection, monitoring and surveillance technologies, especially when human operation is difficult [1,2,3]. In order to track sensor nodes and to apply location-based routing algorithms, developing a localization scheme that takes environment characteristics into account is required. Unlike terrestrial WSNs, underwater sensors are often deployed with a large number over a region of interest. To maintain low cost, only a small fraction of sensors are location-aware, while the rest of them need to be located. Underwater systems inevitably have limited capabilities in storage and computational, fundamental principles for designing localization schemes are rapidity, scalability, and reliability. In the end, the design of localization algorithms should depend on several factors including accuracy requirement, resource ability, and deployment restriction.

In this paper, we study a three-dimensional (3D) scenario and design a received signal strength (RSS) localization scheme for underwater sensor networks using acoustic signals. In fact, due to the high electromagnetic wave attenuation in underwater environments, acoustic channels are mainly employed in UWSNs; thus, the obtained ranges using acoustic signals are more accurate than common radio [2]. For system settings, some special nodes (e.g., anchor nodes, beacon nodes) meaure their global locations. They are often deployed on water surfaces and can cooperatively determine their locations by sharing, while normal nodes (e.g., sensors) have unknown locations. A normal node can obtain its depth information in the water by a pressure sensor. Based on the fact that radio signal strength decreases during propagation, the RSS-based localization process refers to the procedure of determining a sensor location based on the RSS information provided by a number of anchor nodes. Due to RSS propagation factors, several RSS-based localization problems have been addressed [2,3]. Since RSS is easy to collect and highly dynamic in environmental changes, it provides a cheap and fast solution for obtaining the relative distance of sensor nodes without any further computation. On the other hand, it may contain large noise that needs to be refined during the localization process.

RSS-localization has attracted a lot of efforts from different literature, especially in 3D UWSNs. Under an assumption that range measurements are corrupted by the Gaussian noises, a maximum likelihood estimator (MLE) [4] was studied for UWSNs in which a search-based method was adopted. As the maximum likelihood function is multi-modal, it requires a sufficiently close initial guess at the beginning of sequence to obtain a good global solution. Another approach [5] provided an incomplete matrix optimization method by recovering missing entries of a Euclidean squared distance matrix; thus, the global solution was guaranteed. However, the approach [5] may be impractical for huge-size UWSNs because its cost function is nonlinear and the computational burden appears during Newton’s iteration. A convex approximation localization [6] was proposed by applying a projection onto a convex set. In particular, the authors [6] found an appropriate projection onto several numbers of circles to approximate the potential of sensor location, then the estimated location must be located in the intersection of the circles. While [6] has low complexity and reduces the system implementation costs, it cannot adapt to the mobility and dynamic changes in UWSNs. With the weighting factor of nearest reference points [7,8], the authors first derived the MLE for estimating the sensor location, where the transmit power can be treated as either a known or unknown parameter, applied several approximations to convert the MLE into an easy solvable form, then performed a generalized trust range region method. One big disadvantage of the approach is that the localization performance may be degraded through several approximations, and the obtained solution may be a sub-optimal or local one. In terms of energy-based localization, other approaches such as weighted least squares (WLS) were proposed in several works [9,10,11], which offer close-form solutions and low complexity levels. While the authors [10] only considered a simple underwater acoustic transmission path loss model with medium parameters, authors in [11] presented a distributed localization scheme with periodic beacon signals, where an additional time synchronization protocol must be executed before performing the localization process. It has been shown in [10] that the localization scheme was accurate and less vulnerable to environments compared to the time-of-arrival-based scheme [11]. The requirement of time synchronization can also be neglected.

As we introduce above, the conventional localization algorithms have a same common weaknesses. First, they all reply on the assumption of low noise conditions, and thus, cannot adapt to noise uncertainty. If the environment is noisy, their localization performances are degraded due to imprecise location information. Second, randomness in anchor locations is ignored, i.e., they are assumed to be fixed, while in practice, their positions move according to wave motion, wind, etc. In order to overcome those issues, we make three key modifications in our proposed localization approach: noise covariance in RSS measurements, random deployment for anchor nodes, and a linear regression model. For instance, we investigate the covariance among RSS measurements obtained by sensors. The position of the anchor node is assumed to be randomly distributed in an area of interest. We also develop a linear regression model that allows us to access a quick range of information despite of noise condition. This work is a further development of the RSS-based WLS technique [12], where the contributions of this paper are summarized as follows.

(i)We first establish an RSS-based underwater acoustic localization problem in 3D-UWSNs. A set of anchor nodes is randomly deployed on the surface of water and other normal sensors are deployed underwater in a 3D space. The question is how to estimate the normal sensor locations with RSS measurements from anchor nodes, while achieving a high localization performance, short response time and low energy consumption. In order to reduce the 3D localization problem to the 2D space, each sensor is equipped with an inexpensive pressure sensor which can quickly calculate the sensor depth.(ii)We develop a localization scheme for locating the dynamic nodes in a 3D UWSN. First, we suppose there are *N* reference points available and calculate the relevant parameters such as transmit power and path loss exponent by using the least squares method. Thus, several candidate range estimations can be obtained according to the previous subset of samples. Multilateration takes all distance estimations as the inputs and locates the sensor node in the next step. In order to be computationally efficient and easy to implement, we build a linear regression model for obtaining a direct distance ranging in which a given RSS value is further converted to the distance estimate. Moreover, after each multilateration, we check the consistency of range estimation by taking a relational distance adjustment.(iii)In the simulation results, we compare our work with other existing approaches in terms of localization accuracy. To be specific, it is defined by the probability of successful localization, which indicates the ratio of the localization results satisfying a predefined accuracy requirement. Through several case studies, we show that our localization scheme outperforms others in some scenarios.

The remainder of the paper is organized as follows. In Section 2, we present a 3D UWSN system including system architecture and a wireless signal propagation model. In Section 3, we present our problem formulation and the proposed localization scheme for improving the location accuracy and reducing response time. Simulation results are demonstrated in Section 4, followed by conclusions in Section 5.

## 2. Overview

The aim of this section is to first introduce a 3D UWSN system that collects data in a centralized manner, then to present those factors which affect the signal propagation. In order to utilize RSS data collection, we investigate a practical wireless propagation model that reflects the environmental characteristics. For a set of potential anchor nodes, a stable relative RSS distribution over the spatial space reveals high accuracy location of the sensor.

### 2.1. Notations

The following notations are used in the paper.

${\left(\cdot \right)}^{T}$ and ${\left(\cdot \right)}^{-1}$ denote the transpose and inverse operators, respectively.For a given random variable *X*, E(X) and Var(X) correspondingly represent the expectation and the variance of *X*. Pr(X≤x) means the probability of which *X* is less than a value *x*.If *X* follows a normally distribution with mean μ and variance $\sigma^{2}$, we write $X∼N(\mu ,\sigma^{2})$.If *X* has an exponential distribution with a parameter λ>0, i.e., X∼Exp(λ), its probability density function is given by $f_{X}\left(x\right)=\lambda e^{-\lambda x}$ for x≥0.

### 2.2. System Model

Figure 1 depicts a deployment scenario for a 3D UWSN. In this system, the network consists of three main components: anchor nodes, normal nodes (or sensors), and a base station (BS). Generally, the anchor nodes can be deployed on a sea surface like buoys, which float according to the motion of water flows and waves, but are attached by a rope to the seabed. Their locations are often known because they are equipped with global positioning system (GPS) modules. The normal sensors are randomly or uniformly deployed with different depths to cover the target area. They are in passive motion with the surrounding to observe and transmit data to the surface anchor nodes. The anchor node collects measurements from the sensors and delivers to the BS where the localization algorithm and further data analysis are performed. Thus, the BS has high power and large capacity for computing and storage.

In order to reduce the computational complexity of localization task as the number of sensors grows, each sensor has ability to retrieve its depth information by being equipped with a pressure sensor. From theoretical perspective, a simple relationship between the pressure $F_{total}$ and the sensor depth *h* is given as
(1)$$F_{total}=F_{atmosphere}+\left(\rho \times g\times h\right),$$
where $F_{atmosphere}$ is the atmospheric pressure, ρ is the water density, which varies according to the type of underwater conditions (e.g., deep sea, clean ocean, or coast), and *g* is the acceleration of gravity. From a practical perspective, several manufacturers have produced pressure instrumentation with high accuracy and adaptability to harsh environments. For instance, Paroscientific Digiquartz’s depth sensors [13] can calculate pressure and achieve a highly accurate depth measurement (about 99.9%) up to 7000 m water depth. Thus, errors that come from depth measurements are neglected in this paper.

Regarding to transmission strategies, existing mechanisms of data transmission process can be divided into two main categories: demand-driven and event-driven [11]. While a sensor answers the requests by sending back a message (which contains the required contents) to the BS in the demand-driven-based transmission, it only reports when it perceives a target event in the event-driven mechanism. When a sensor initially detects the object, data packets generated by the sensor are sent towards to the anchor nodes on the surface through a distributed manner. A data packet contains the sensor’s identity, the recorded RSS measurement, and the depth information. Until the packet reaches the surface anchor nodes, they collect and deliver back to the BS with their locations. Finally, a centralized localization algorithm can be executed by the BS to obtain the sensor location based on the received packets.

### 2.3. Wireless Signal Propagation Model

Due to the limitation in the storage and processing capacity of sensors, developing a less complex measurement method for them is an important issue. RSS measurement is a commonly known method, which is used for measuring signal strength by a receiver. It not only offers a low cost and easy implementation solution but also allows us to obtain a proximity of neighboring range instead of direct-distance measurement. Thus, several works have investigated the RSS-based localization problems for many location-aware applications. Under the system architecture in Section 2.2, we aim to formulate the localization problem in terms of RSS. Mathematically, for a given distance *d* between two sensors, RSS reading is defined by the observed path loss at $d\geq d_{0}$ as [14,15]
(2)$$P\left(d\right)=P\left(d_{0}\right)-10\gamma \log_{10}\left(\frac{d}{d_{0}}\right)-A\left(d,f\right)+X_{\sigma }\;\left[\mathrm{dB}\right],$$
where $P(d_{0})$ represents the signal power loss at the reference distance $d_{0}$, γ is the path loss component, and $X_{\sigma }∼N\left(0,\sigma^{2}\right)$ is the shadow fading. Equivalently, P(d) can be expressed as $P\left(d\right)=P_{t}-P_{r}\left(d\right)$, where $P_{t}$ and $P_{r}\left(d\right)$ are the transmit signal power and the received signal power at the distance *d*. The attenuation A(d,f) is calculated as a function of the distance *d* and the frequency *f* according to the Thorp model [7,8].
(3)$$A\left(d,f\right)=10k\log_{10}d+d\zeta \left(f\right).$$

Here, k(1≤k≤2) is the geometrical spreading coefficient (e.g., k=1 and k=2 represent the cylindrical and spherical geometries, respectively), ζ(f) is a frequency-dependent absorption coefficient given as
(4)$$\zeta \left(f\right)=\frac{0.11f^{2}}{1+f^{2}}+\frac{44f^{2}}{4100+f^{2}}+2.75\times 10^{-4}+0.03\;\left[\mathrm{dB}/\mathrm{km}\right].$$

The model in (Equation 3) is often applied to characterize the acoustic channel in underwater communications. This representation is very simple to implement and only depends on the signal frequency, thus, other parameters such as water temperature, salinity and acidity are neglected in this paper. Thus, the total signal attenuation at the receiver is obtained by combining the effects of path loss due to absorption and spreading loss. Regarding the fading effects $X_{\sigma }$, we also assume that the correlation between $X_{\sigma }\left(i\right)$ and $X_{\sigma }\left(j\right)$ satisfies
(5)$$E\left[X_{\sigma }\left(i\right)X_{\sigma }\left(j\right)\right]=\sigma e^{-d_{ij}/c},$$
where $d_{ij}$ is the distance between the *i*-th anchor and the *j*-th sensor and *c* is a predefined decorrelation parameter [9]. Note that γ and $\sigma^{2}$ are unknown parameters and can be estimated based on a certain number of the RSS measurement samples.

## 3. Problem Formulation and Solution

In this section, we establish a two-step localization scheme which extracts RSS measurements. First, in order to explore the closeness information on the RSS measurements, two biasing factors (i.e., transmit power and path loss exponent) are calculated according to a given measured set. With the help of this step, an unknown sensor is able to obtain an estimate for its relative distance. Then, for a given number of available anchor references, we can determine the sensor location by applying a multilateration technique. In many cases, since RSS may suffer from noisy ranging measurements, we also develop a consistency check scheme in which the revised distance estimate is adjusted based on the geometrical configuration. Thus, we get a better location estimation with guaranteed accuracy.

### 3.1. Estimation of Unknown Path Loss Factors

If γ and $\sigma^{2}$ are perfectly known and the sensing time is not limited, we can achieve a full knowledge of channel information, thus can obtain an MLE at a certain distance as in [8]. However, these assumptions hardly hold in practical situations. First, since the noise uncertainty is randomly changed in water environments, it is difficult to obtain the exact value of $\sigma^{2}$ at a particular time. Second, time dedicated to the sensing depends on the quality-of-services of the system. From those perspectives, we realize that one of the critical limitations is the issue of matching the observation with the learned elements from environments. It can be solved by either designing a test for hypothesis verification or extracting the characterizing features from the raw sensor data. As shown in the work [12], γ and $\sigma^{2}$ can be evaluated from persistent learning of the area.

Consider a set of *N* observations $P(d_{i})$ at different distance $d_{i}$ from a transmitter, we collect *N* measurement pairs $\left\{P\left(d_{i}\right),d_{i}\right\}$ for the area of interest. The log-likelihood function is obtained from the joint probability density function of *N* measurements as
(6)$$\begin{array}{cc} L(\gamma ,\sigma^{2}) & =-N\ln\left(\sqrt{2\pi \sigma^{2}}\right)-\frac{1}{2\sigma^{2}}\sum_{i=1}^{N}{\left[P\left(d_{i}\right)-P\left(d_{0}\right)+A\left(d,f\right)+10\gamma \log_{10}\left(\frac{d_{i}}{d_{0}}\right)\right]}^{2}. \end{array}$$

Taking the derivative of $L(\gamma ,\sigma^{2})$ respect to $\gamma ,\sigma^{2}$ and compensating to zero, we obtain
(7)$$\begin{array}{c} \hat{\gamma }=\frac{\sum_{i=1}^{N}\log_{10}\left(\frac{d_{i}}{d_{0}}\right)\left[P\left(d_{i}\right)-P\left(d_{0}\right)+A\left(d,f\right)\right]}{10\sum_{i=1}^{N}{\left[\log_{10}\left(\frac{d_{i}}{d_{0}}\right)\right]}^{2}}. \end{array}$$
(8)$$\begin{array}{c} {\hat{\sigma }}^{2}=\frac{1}{N}\sum_{i=1}^{N}{\left[P\left(d_{i}\right)-P\left(d_{0}\right)+A\left(d,f\right)+10\hat{\gamma }\log_{10}\left(\frac{d_{i}}{d_{0}}\right)\right]}^{2}. \end{array}$$

Thus, once those channel parameters are obtained, the distance estimation from (Equation 2) becomes relatively straightforward.

### 3.2. Estimation of Distance and Location

Substituting (Equation 7) and (Equation 8) into (Equation 6), the log-likelihood function becomes a function of single distance *d*, i.e.,
(9)$$\begin{array}{c} L\left(d;\hat{\gamma },{\hat{\sigma }}^{2}\right)=-\ln\left(\sqrt{2\pi {\hat{\sigma }}^{2}}\right)-\frac{1}{2{\hat{\sigma }}^{2}}{\left[P\left(d\right)-P\left(d_{0}\right)+A\left(d,f\right)+10\hat{\gamma }\log_{10}\left(\frac{d}{d_{0}}\right)\right]}^{2}. \end{array}$$

With the same technique, the measured distance between two sensors can be derived by taking the derivative of $L(d;\hat{\gamma },{\hat{\sigma }}^{2})$ respect to *d* and letting it to zero, which satisfies the following.
(10)$$10\left(\hat{\gamma }+k\right)\log_{10}\hat{d}+\hat{d}\zeta_{f}=P\left(d_{0}\right)-P\left(d\right)+10\hat{\gamma }\log_{10}d_{0}.$$

For solving (Equation 10), an estimated distance $\hat{d}$ is obtained by implementing the graph method [16]. This numerical solution provides an approximation of the corresponding distance between the anchor node and the normal sensor. Then, we can apply a range-based geometric multilateration technique to produce a single location estimation. In particular, the estimated location can be found as the centroid of the intersection of the areas generated by the estimated distance $\hat{d}$ from several anchor nodes.

In the next step, we discuss how to obtain the location of sensor from a set of the estimated distances (Equation 10). Note that an unknown sensor can use the received information to obtained its own location for achieving a course-grained localization. The basic idea of this procedure is the multilateration method. We denote a set of $M_{a}$ reference anchors with the locations $\left\{a_{1},\cdots ,a_{M_{a}}\right\}\;\left(a_{i}={\left[x_{i},y_{i},0\right]}^{T},i=1,\cdots ,M_{a}\right)$ and the location of a normal sensor $x={\left[x,y,z\right]}^{T}$. Our goal is to estimate the sensor location x from the RSS measurements from $M_{a}$ anchor nodes. In general, the multilateration procedure requires at least $M_{a}=3$ in 2D space, and $M_{a}=4$ in 3D space. According to Section 2.2, each normal sensor uses its measured pressure to obtain the depth information. For the sensor x, let *h* denote the corresponding estimated depth value adopting from the pressure sensor and define $z=h,z_{i}=0,R_{i}^{2}={\hat{d}}_{i}^{2}-h^{2}$, we have
(11)$$\begin{array}{cc} \left(\hat{x},\hat{y}\right) & =\arg\min_{\left(x,y\right)}\sum_{i=1}^{M_{a}}\left[{\left(x-x_{i}\right)}^{2}+{\left(y-y_{i}\right)}^{2}+{\left(z-z_{i}\right)}^{2}-{\hat{d}}_{i}^{2}\right]\\ & =\arg\min_{\left(x,y\right)}\sum_{i=1}^{M_{a}}\left[{\left(x-x_{i}\right)}^{2}+{\left(y-y_{i}\right)}^{2}-R_{i}^{2}\right], \end{array}$$

In another expression, the 3D localization problem was reduced to a 2D one. From (Equation 11), the sensor location can be obtained by solving Ax=b, which leads to a least-square solution $\hat{x}=\frac{1}{2}{\left(A^{T}A\right)}^{-1}A^{T}b$. In particular, we have
$$A=\left[\begin{array}{cc} x_{1}-\frac{1}{M_{a}}\sum_{i=1}^{M_{a}}x_{i} & y_{1}-\sum_{i=1}^{M_{a}}y_{i}\\ \vdots & \vdots \\ x_{M_{a}}-\frac{1}{M_{a}}\sum_{i=1}^{M_{a}}x_{i} & y_{M_{a}}-\sum_{i=1}^{M_{a}}y_{i} \end{array}\right],$$
$$b=\left[\begin{array}{c} \left(x_{1}^{2}-\frac{1}{M_{a}}\sum_{i=1}^{M_{a}}x_{i}^{2}\right)+\left(y_{1}^{2}-\sum_{i=1}^{M_{a}}y_{i}^{2}\right)-\left(R_{1}^{2}-\sum_{i=1}^{M_{a}}R_{i}^{2}\right)\\ \vdots \\ \left(x_{M_{a}}^{2}-\frac{1}{M_{a}}\sum_{i=1}^{M_{a}}x_{i}^{2}\right)+\left(y_{M_{a}}^{2}-\sum_{i=1}^{M_{a}}y_{i}^{2}\right)-\left(R_{M_{a}}^{2}-\sum_{i=1}^{M_{a}}R_{i}^{2}\right) \end{array}\right].$$

This procedure will produce a coarse solution after obtaining enough number of anchor nodes. Finally, it takes $M\times \left(\frac{M-1}{2}\right)$ equations needed to solve. This number of equations increases substantially as the number of anchor nodes used for the localization increases.

### 3.3. Regression Model

Unlike conventional approaches, there are three main challenges at the BS to obtain the sensor location from its received packets, which are listed as follows.
(i)First, RSS measurements depend on the unknown location x and channel factors, thus are time-varying at the BS. Thus, it takes more than three reference anchor nodes for the BS to calculate the exact solution by only using (Equation 2).(ii)Second, RSS is typically affected by shadowing and multipath effects. If we learn a set of anchor nodes relatively stable over the space, the measured RSS can reveal the exact location of the unknown sensor.(iii)Third, it is challenging for the BS to estimate all the above parameters and to response in a short time. In fact, the localization task should be fast and energy efficient. While the sensor only broadcasts a localization request message when it detects an event, the BS should proceed with the message soon before the sensor may be changed due to water motion.

In order to solve those aforementioned problems, we need to collect the distribution of the RSS of the anchor nodes to conduct a site survey in the area by recording the RSS measurements at each reference location. By this way, a linear regression model is proposed to responsively obtain a relationship between the RSS measurements towards the distance. In particular, denoting the ideal distance as $d_{i}$, the polynomial fitting function for the set of *N* points $\left\{d_{i},P_{i}\right\}$ is defined by
(12)$$d_{i}\left(P_{i}\right)=\alpha_{0}+\alpha_{1}P_{i}+\cdots +\alpha_{n}P_{i}^{n},$$
where $\alpha_{j}\in R\;\left(j=0,\cdots ,n\right)$ are the coefficients and *n* is order of the polynomial. Given a sequence of distances between two sensors ${\hat{d}}_{i}$, we suppose that each distance is determined on a predefined interval. According to the Weierstrass approximation theorem [17], for each tolerance ϵ>0 there exists a polynomial $d_{i}\left(P_{i}\right)$ of an appropriate order, which approximates ${\hat{d}}_{i}$ within the given tolerance, i.e.,
(13)$$|{\hat{d}}_{i}-d_{i}\left(P_{i}\right)|\leq \epsilon .$$

Given ϵ, we need to calculate the coefficient subject to minimizing the sum of square errors. In particular, let $\varepsilon \left(a_{0},\cdots ,a_{n}\right)=\sum_{i=1}^{N}\varepsilon_{i}^{2}$, where $\varepsilon_{i}=\left({\hat{d}}_{i}-d_{i}\right)$, we write
(14)$$\left(\alpha_{0}^{*},\cdots ,\alpha_{n}^{*}\right)=\underset{\left(\alpha_{0},\cdots ,\alpha_{n}\right)}{\arg\min}\;\varepsilon =\underset{\left(\alpha_{0},\cdots ,\alpha_{n}\right)}{\arg\min}\sum_{i=1}^{N}{\left({\hat{d}}_{i}-d_{i}\right)}^{2}.$$

Taking partial derivatives of ε with respect to $\alpha_{0},\cdots ,\alpha_{n}$ and letting them approach zero yields to
(15)$$\begin{array}{cc} \frac{\partial \varepsilon }{\partial \alpha_{j}}=0 & \Leftrightarrow \sum_{i=1}^{N}\left(-2P_{i}^{j}\right)\left({\hat{d}}_{i}-d_{i}\right)=0\\ & \Leftrightarrow \sum_{i=1}^{N}\left(-2P_{i}^{j}\right)\left({\hat{d}}_{i}-\alpha_{0}-\alpha_{1}P_{i}-\cdots -\alpha_{n}P_{i}^{n}\right)=0\\ & \Leftrightarrow \sum_{i=1}^{N}P_{i}^{j}{\hat{d}}_{i}=\alpha_{0}\sum_{i=1}^{N}P_{i}^{j}+\alpha_{1}\sum_{i=1}^{N}P_{i}^{j+1}+\cdots +\alpha_{n}\sum_{i=1}^{N}P_{i}^{j+n}, \end{array}$$
for all j=1,⋯,n. By combining (Equation 12) and (Equation 15), the regression coefficients $\left(\alpha_{0}^{*},\cdots ,\alpha_{n}^{*}\right)$ can be numerically found. The linear regression model has many good properties. From the standpoint of the execution time, it is capable of being flexibility for non-complicated data and, overall, the fitting process is simple. Furthermore, it still performs well even when the error distribution associated with the measurement data is not normal, which are important for stable localizability. It also shows a precise way to transform an RSS value to a distance estimate with a given upper bound of errors. In Figure 2, we provide four examples of polynomial regression fits (PRFs) in several orders. In this figure, the solid black line interpolating circles and the curve of black square dots represent the ideal and real RSS versus distance, respectively. The fitting polynomials (Equation 12) in several orders are plotted with different colors. For instance, the first order PRF is specified by a dashed curve with red cross markers. We also add a text description (in red color) on the average root mean square error (ARMSE) between the real distance and the estimated one via fitting for each case. We observe that with higher PRF more accurate distance translation is obtained, thus we get a better location estimation.

### 3.4. Adjustment of Relational Ranging

According to the previous analysis, the RSS measurements are often affected by the noise and interference in the environments. Thus, in this section we propose a sufficient and necessary method to reduce those phenomena that cause errors in distance estimation. The basic idea is that each normal sensor requires its location in a certain time defined by a number of rounds. At the beginning of each round, the BS estimates the sensor location by computing a least-squares-based multilateration solution. In general, the relative distances from sensor to anchor nodes will be refined according to an error constraint controlled by the BS. The distance adjustment is done after achieving a desired accuracy or a specific number of iterations. Without loss of generality, suppose that an estimated distance $\bar{d}$ can be expressed as a summation of the true distance *d* and the error δ as
(16)$$\bar{d}=d+\delta_{d}.$$

Here, the error $\delta_{d}$ proportional to *d*, i.e., $\delta_{d}=\xi d$, where ξ is a random variable with mean E(ξ)=η. This is a clear definition for two reasons. First, since the aim of this section is to reduce the uncertainty of distance estimation in the previous localization process, it is more important to ignore other factors. Second, although we may not know how much ξ is proportional to *d*, adjusting its value as a random parameter still guarantees the fairness among random factors in distance uncertainty. As soon as the distribution of parameter η is known, the refine distance can be readily computed as
(17)$$\tilde{d}=\frac{\bar{d}}{1+\beta }\;,$$
where β is chosen to minimize the mean square error $E[{\left(\tilde{d}-d\right)}^{2}]$. Here, we ignore the subindex of distance value for the simplicity purpose. The basic idea is that for a random subset of variable ξ, we use it to compute several candidate estimations $\tilde{d}$ according to (Equation 17). Then, the one with the least mean square error is chosen as a tentative estimate. The term $E[{\left(\tilde{d}-d\right)}^{2}]$ can be expressed as
(18)$$E\left[{\left(\tilde{d}-d\right)}^{2}\right]=E\left[{\left(\frac{\bar{d}}{1+\xi }-\frac{\bar{d}}{1+\beta }\right)}^{2}\right].$$

Thus, results from taking derivative on both sides of (Equation 18) and setting it to zero, we obtain
(19)$$\frac{d}{d\beta }E\left[{\left(\frac{\bar{d}}{1+\xi }-\frac{\bar{d}}{1+\beta }\right)}^{2}\right]=E\left[\frac{1}{1+\eta }-\frac{1}{1+\beta }\right]=0,$$
which yields to $\beta =E{\left[{\left(1+\xi \right)}^{-1}\right]}^{-1}-1$. For example, we consider the case that ξ follows the exponential distribution, i.e., ξ∼Exp(λ), where $\lambda =\frac{1}{\eta }$. Thus, the value β can be calculated as
(20)$$\begin{array}{cc} \beta & ={\left[\int_{0}^{\infty }\frac{1}{1+\xi }\eta e^{-\eta \xi }d\xi \right]}^{-1}-1={\left[\frac{1}{\eta }e^{\frac{1}{\eta }}E_{1}\left(\frac{1}{\eta }\right)\right]}^{-1}-1, \end{array}$$
where $E_{1}\left(u\right)=\int_{u}^{\infty }e^{-t}/tdt$ is the special exponential integral. Figure 3 shows an example of distance adjustment following the above procedures. The simulation parameters are given in Section 4. In Figure 3a, the vertical and the horizontal axes represent the corresponding total lengths of the distances that we perform. Due to the mutually perpendicular plane property in 3D space, the distance between two points in the space can be defined by this way. In each run, the coordinates of a single sensor were randomly generated from a uniform distribution. The estimated range was obtained from the raw measurements (Equation 2) by applying the dead-reckoning method [3]. Thus, the adjusted distances show the number of runs in which the optimal solution was given by (Equation 17) (averaged over 50 realizations). Note that for high noise level, the optimal adjusted distances have a decent performance because the estimated distances from the raw data are unstable and have the huge norm, which causes their averages to be large. Also, Figure 3b shows the average error during the refinement phase versus given error mean η of the random variable ξ∼Exp(1/η). In this figure, we keep a fixed real distance d=50 m, generate its estimated distance according to (Equation 16), and compute the corresponding refined distance by using (Equation 17). We observe that the average error increases as η increases. Obviously, a larger region means less precision and lower quality of distance estimation.

### 3.5. Data Transmission Strategy for Localization Task

It is obvious that our data transmission strategy follows an event-driven mechanism. For a given network, a node is called $N_{h}$-hops localizable if it can be localized by using only measurement information at most $N_{h}$-hop references. In particular, the anchor nodes cooperatively estimate the per-hop distance, i.e., the *i*-th anchor node calculates the per-hop distance and stores itself as
(21)$${\mathrm{Ph}}_{i}=\frac{\sum_{i\neq j}\left|\right|a_{i}-a_{j}\left|\right|}{\sum_{i\neq j}h_{i}}\;\left(i,j=1,\cdots ,N\right),$$
where $h_{i}$ is the number of hops from the *i*-th anchor node to the *j*-th one. We denote $N_{h}^{\left(i\right)}$ as the minimum hop count between the unknown node and the *i*-th anchor node, then the distance between them can be roughly estimated by
(22)$${\bar{d}}_{i}=N_{h}^{\left(i\right)}\times {\mathrm{Ph}}_{i}.$$

If the value $N_{h}^{\left(i\right)}=1$, the distance estimation ${\bar{d}}_{i}$ can be further simplified as (Equation 12). In the following, we present a distributed protocol to locate a sensor node in the network. This protocol is an interactive manner in which a localization task follows step-by-step procedures and stops when the BS obtains the sensor location.

We give some comments on Algorithm 1. Regarding the location request message, it contains the identity of the normal sensor, its RSS record to its neighbor surface anchor, the depth information, and the hop count to reach the anchor nodes. Hence, before sending the message, each normal sensor estimates the least hop counts to each anchor node. In order to achieve a desire localization result, we define a predefined threshold for the multilateration error. Localizability information diffuses step-by-step paths following Section 3.2 and stops when the localization error reaches the threshold or after a number of predefined iterations. In terms of energy saving, since sensor node only broadcasts the packet when it senses an event, this data transmission mechanism can reduce the signaling cost and the number of sending packets, while the BS has large storage and computational capability to obtain more accurate results and to reduce the overheads over the sensors.  

**Algorithm 1** A framework of data transmission
1:A normal sensor senses an event, it broadcasts a localization request message towards the neighbor anchor nodes over a period.2:Surface anchor nodes receive the packet, they associate their locations with the message and forward to the BS3:When the anchor’s message reaches the BS, the distance $\bar{d}$ is calculated according to (Equation 12) and is stored at the BS.4:The BS updates *M* entries of $\bar{d}$ in the localization table and uses them for obtaining sensor location according to Section 3.2 and adjusting the distance value according to Section 3.4.


## 4. Simulation Results

In this section, we aim to conduct several numerical simulations in order to assess the effectiveness of the proposed localization scheme. Regarding the simulation parameter settings, we consider a networks of $N_{node}$ nodes that are randomly deployed in a seabed of 1000 × 1000 × 600 m with the same amount of energy. The minimum distance between the sensors and the minimum distance between the anchor nodes are 36 m and 70 m, respectively. Other parameters are given in Table 1. The sensors are deployed in different depths of water. We also assume that the distance between normal sensors and the surface anchor nodes should meet the connectivity requirement.

For example, Figure 4 shows a sensor deployment scenario given anchor nodes and normal nodes. Our localization scheme is also based on several assumptions as follows. First, we generate a network of $N_{node}$ randomly and uniformly deployed nodes in the predefined area of interest in which the communication radius is adopted from Table 1. Anchor nodes are randomly distributed on the water surface in which the distances among them should be equal to or greater than the preset parameter and guarantee a cover of the surface of the seabed. Sensor nodes are deployed at various depths, thus, they can report an event to the surface anchor nodes by a single hop or multiple hops.

Second, an anchor node can obtain its global location either by using a GPS module or sharing with the local neighboring nodes. In practice, the location measurement inevitably contains errors, resulting in an inaccurate solution. We suppose that the effect of errors can be neglected, thus, the multlateration technique can be applied on (Equation 11). For the movement trajectory of an anchor node $a={\left[x_{a},y_{a},0\right]}^{T}$, we adopt the noise model used in [18] to represent the noises in processing motion and measurement phases. In this study, the noise is modeled as a function of time *t* as
(23)φ(t)=1.2+0.3cos(0.4t).

The change of the anchor coordinate coincides with the Bower’s function $\Phi =\Phi (x_{a},y_{a},\varphi )$ [18].
(24)$$\delta_{x_{a}}=-\frac{\partial \Phi }{\partial y_{a}};\;\delta_{y_{a}}=\frac{\partial \Phi }{\partial x_{a}}.$$

We also keep the same set of related parameters of [18]. Thus, the coordinate of the anchor node at a certain time *t* is calibrated according to the previous coordinate and its motion change.
(25)$$a^{t}=\Phi \left(x_{a}^{t},y_{a}^{t},\varphi^{t}\right)\approx a^{t-1}+\Delta \Phi$$
where $\Delta \Phi ={\left[\delta_{x_{a}},\delta_{y_{a}},0\right]}^{T}$. Note that we only present a simple model for modeling the anchor movement, but many complex observation models have been investigated to eliminate inconsistencies and inaccurate data, which is out of the scope of this paper. Third, in order to reduce hardware cost and to preserve network energy, only a small fraction of the nodes is location-aware. In this case, anchor ratio (AR) is defined by the number of anchor nodes over the total number of sensors in the network. Thus, when alternative anchor references are available, any subgroup can locate a normal sensor by the multilateration to provide a desire result. Note that the AR can be selected according to the quality performance metrics. It is often desired to be as low as possible. Thus, the AR is the minimum number that guarantees a high accuracy given an acceptable error. Third, in order to accurately estimate the unknown path loss factors in (Equation 7) and (Equation 8), the RSS should be sampled from distinct separate points in space and the number of measurement pairs *N* is sufficient according to the total number of sensors.

In order to evaluate the performance of our proposed localization scheme, we consider the following performance metrics. First, the root mean square error (RMSE) gives us information on the difference between the actual sensor position x and the estimated one $\hat{x}$. The RMSE is evaluated through the Monte Carlo simulations, i.e., for a number of runs $N_{s}$ is defined by
(26)$$\mathrm{RMSE}=\sqrt{\frac{1}{N_{s}}\sum_{i=1}^{N_{s}}||x-\hat{x}{\left|\right|}_{2}^{2}}.$$

In fact, we set the maximum number of repeated random samplings as the number of sensor nodes. Second, for a given upper bound of the localization error ϵ, the corresponding cumulative distribution function (CDF) of the average RMSE is given by
(27)CDF(ϵ)=Pr(RMSE≤ϵ),
where the right-hand side of (Equation 27) represents the probability that RMSE takes on a value less than or equal to ϵ. Moreover, a good localization scheme should achieve a proper localization error in low AR and its performance should be independent on the node density.

Figure 5 first illustrates the attenuation in (Equation 3) versus the frequency measured in dB when the transmission range is the maximum. In the propagation model (Equation 2), there are two terms in the path loss, which are determined by the distance *d*. The term $10\gamma \log_{10}\left(d/d_{0}\right)$ determines the space spread, and the term A(d,f) represents the attenuation coefficient. As a result, the attenuation tends to increase as frequency or distance increase, while the path loss simultaneously decreases in signal power. The attenuation value almost hits zero at the frequency of f=50 kHz. The figure it provides us information on path loss-distance characteristics of the area. Hence, it is important to associate collected data with locations for making the data set meaningful.

Figure 6a plots the probability of localization success at different water depth *D*. We observe that the probability of localization success increases as either *D* increases or AR increases. This is because of two possible reasons. First, as *D* increases, the value A(d,f) becomes small and, thus, can be ignored in the depth water. The sensor’s movement in the deep sea is smaller than that in the shallow sea. Moreover, it has been shown that localization with multi-hop transmission can prevent the propagation of localization errors and require low communication overheads [5]. Since we fixed M=3 in the simulation settings, we may increase the value of *M* in the multilateration procedure to achieve a better performance. Thus, it needs to establish a specific upper bound for *M* in order to balance the tradeoff between the computational cost and the localization accuracy. One suggestion is the development of a feedback control strategy based on the sensed data, where the knowledge of having a specific value of *M* is not required.

Figure 6b compares the CDF performance for a given upper bound of localization error ϵ. For example, for given ϵ=20 and AR = 3%, the probability that the average RMSE is lower than ϵ is around 0.47. It is shown that increasing the AR will improve the RMSE performance. Based on the results, we can choose an appropriate AR and a suitable upper bound of localization error to get an acceptable RMSE. For instance, when AR = 12% and ϵ=40, more than 85% of the average RMSE value is smaller than ϵ. As we explained above, selecting a small value of AR provides a win–win pricing deployment strategy, while guaranteeing a good RMSE. Thus, this result gives us evidence to choose the appropriate AR for a given localization error bound.

Figure 7 shows performance comparison between the proposed scheme and other methods in terms of (a) the impact of water depth and (b) the influence of node density. In this section, there are two other schemes we would like to compare with, which are second-order cone programming (SOCP) [6] and WLS [7]. The localization ratio versus water depth is shown in Figure 7a. From the simulation results, we observe that our proposed scheme achieves more stable performance in different water depths than the others. As *D* increases, the SOCP scheme has a slightly higher localization ratio, because the geometry relationship among the nodes changes slowly and the attenuation coefficient can be neglected at deeper water depths. We also study in Figure 7b the localization performance in terms of the number of nodes for a given area, where AR is fixed at 10%. The proposed scheme identifies about 9–18% non-localizable nodes, while about 12–21.71% in SCOP and 15.19–30.11% in WLS. This is because the two conventional approaches suffer from the ranging estimator errors that arise from the RSS measurement inaccuracy, while in our approach, the distance is obtained from a learning model, then it is refined to remove the unwanted errors.

**Remark** **1** (Summary of findings)**.**
*This remark ends with a summary to provide a big picture and facilitates understanding. In this paper, we study the RSS localization problem based on an event-driven mechanism for UWSNs. In recent years, this issue has drawn remarkable attention from an increasing number of researchers. A localization task refers to the estimation procedure of an ordinary sensor node in a network. In order to design a good localization, we consider the localization issue based on various properties as the following.*

*Regarding network topology, we consider a geometric model to reflect how a sensor network observes an event over the physical space. In particular, normal sensor nodes are placed at different depths to cover the area of interest. Anchor nodes are deployed on a sea surface and drift with water motions. A network node collects measurements and delivers back to a BS for analysis, thus, it resolves the computation limitations of sensor nodes. It has been shown that this network topology is able to obtain accurate and reliable results. One problem is the communication cost of transmitting data back to the BS. Thus, a good localization strategy should reduce the number of sending packets.*

*In terms of the measurement model, a practical RSS model was considered in (2). Thus, a precise distance was obtained for estimating the unknown sensor node more accurately. In Section 3, we showed how to transform RSS measurements for the location of a normal node. This step typically takes place among the target node and its neighbor anchors. Depending on the number of hop counts $N_{h}$, we name the process as $N_{h}$ localization estimation. Moreover, errors are often inevitable and unpredictable for RSS-based ranging techniques. Thus, the location refinement was proposed in Section 3.3 to improve the accuracy of the localization scheme.*

*Through analyses and simulation results, we address some effectiveness of the proposed scheme as follows. First, as we presented in Section 2.2, since each normal sensor is equipped with a pressure sensor, the 3D localization problem can be reduced to the 2D space. Moreover, by establishing a regression model between RSS–distance for a given upper error bound, we can balance the network complexity and the response time during the localization process. Second, the proposed localization scheme is based on an event-driven mechanism, so that it can save the energy for the sensor nodes because it is not required to be fully awake.*



## 5. Conclusions

This paper has studied an event-driven RSS-based localization in UWSNs. Based on the establish model system from a centralized perspective, we constructed and proposed a fast localization algorithm by exploiting a linear fitting model for an RSS–distance relationship among the sensors. In order to mitigate the impact of errors in distance estimation, we also developed a method for sequentially computing and refining the relative distance. Simulation results show that the localization accuracy has been improved.

Based on the results, we make the following discussions. First, in our problem formulation, we use the Thorp model to simulate the attenuation coefficient factor. This value only depends on the distance and the frequency and, thus, may not incorporate dynamic propagation changes in water depth and random wave motions. Therefore, in order to predict the sensor location more accurately, we need to investigate other nonlinear models that represent the changes in the range, the depth and the form of the ocean waves. In our paper, we obtain the RSS variance and the path loss exponent value by learning from a certain number of samples in the first stage, then, the coordinates of a sensor are estimated iteratively in the second stage. In general, it is necessary to estimate the path loss exponent and the location jointly, therefore, it is legitimate to use the information (Equation 9). Although our localization scheme requires a prior learning process to obtain a good regression model, it returns a better and more stable performance through several case studies. Thus, depending on the purpose of the applications, we can choose the model with the desire accuracy. Moreover, in order to efficiently apply a localization scheme in UWSNs, features and challenges of underwater acoustic signals should be integrated with localization designs.

## Figures and Tables

**Figure 1 sensors-19-03105-f001:**
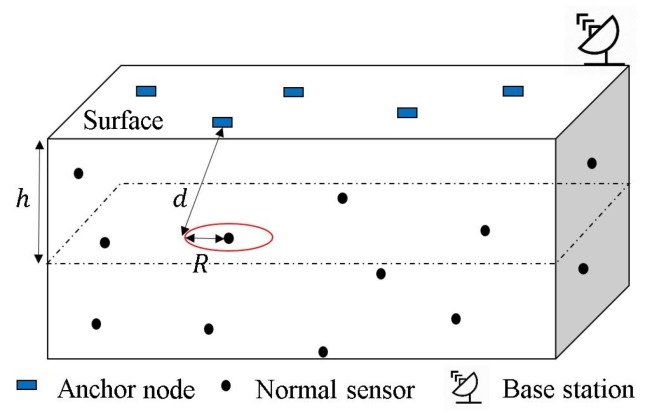
Deployment scenario for 3D localization scheme.

**Figure 2 sensors-19-03105-f002:**
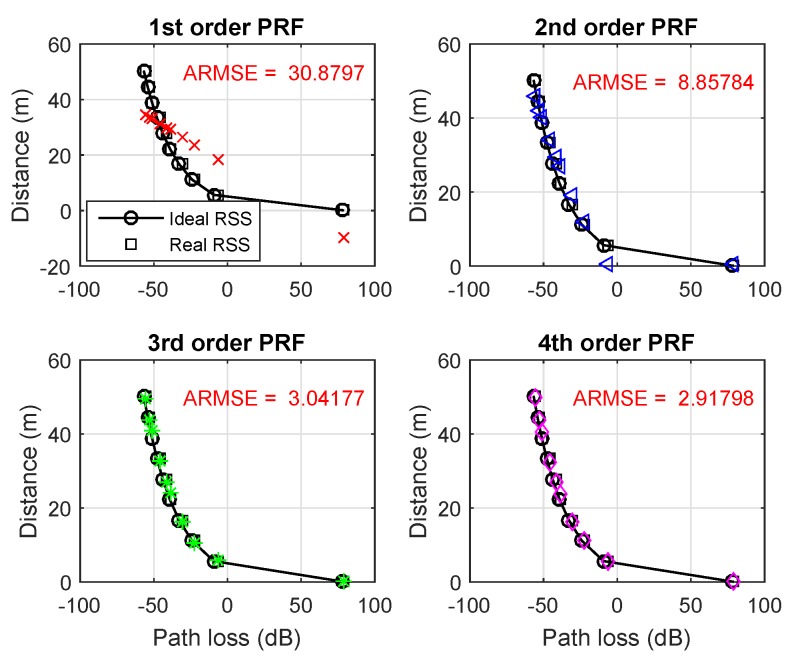
Fitting polynomials for different orders.

**Figure 3 sensors-19-03105-f003:**
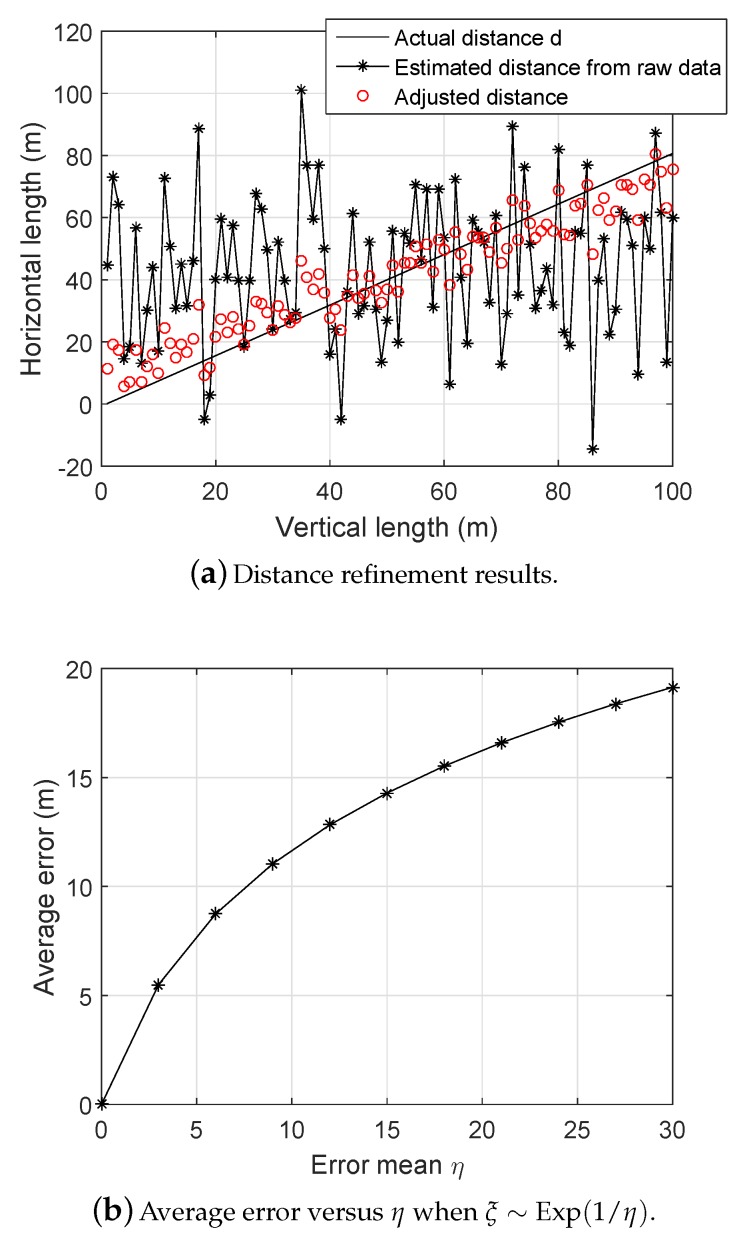
Examples of distance calibration. (**a**) Distance refinement results; (**b**) Average error versus η when ξ∼Exp(1/η).

**Figure 4 sensors-19-03105-f004:**
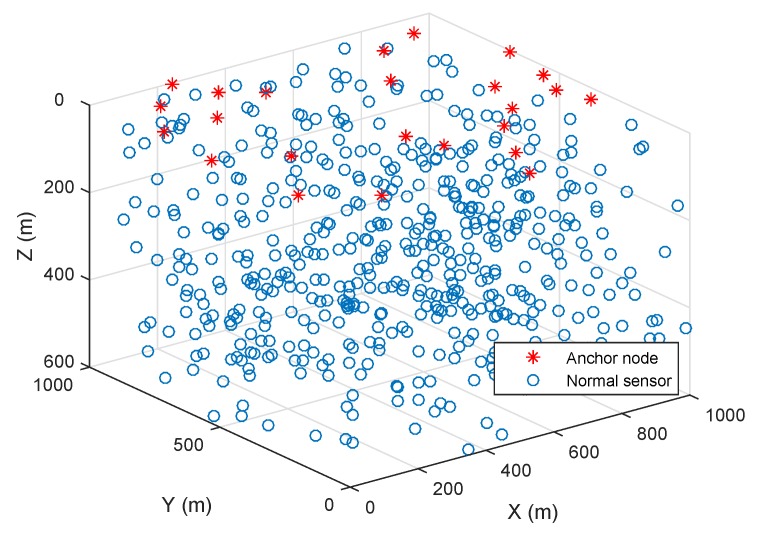
Deployment scenario example.

**Figure 5 sensors-19-03105-f005:**
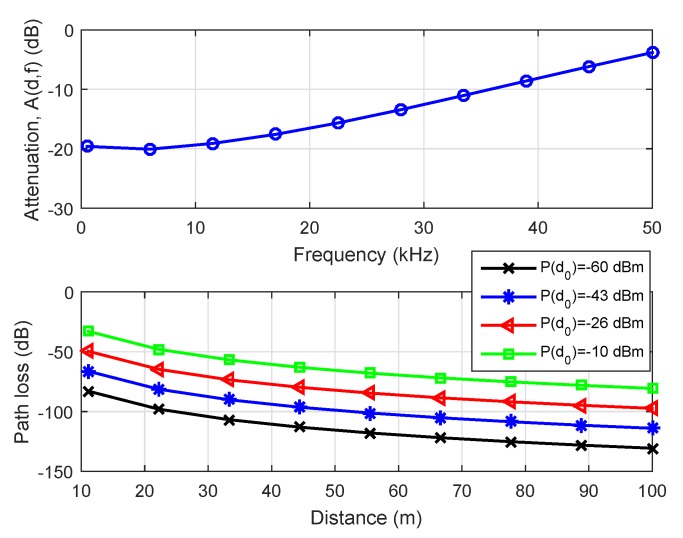
Attenuation coefficient A(d,f) versus frequency (top figure) and path loss versus distance (bottom figure).

**Figure 6 sensors-19-03105-f006:**
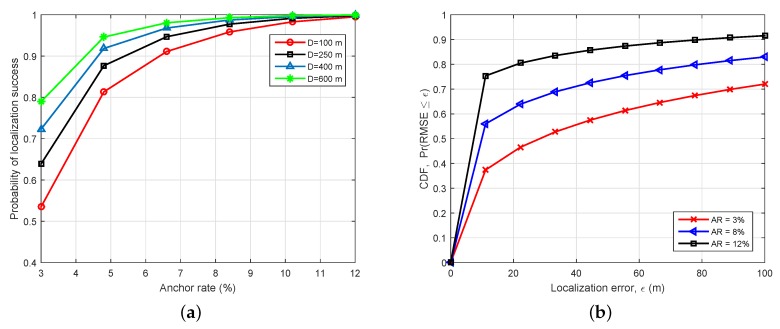
Localization performance of the proposed scheme. (**a**) CDF performance versus AR for several water depths; (**b**) Probability of successful localization versus localization error for several ARs.

**Figure 7 sensors-19-03105-f007:**
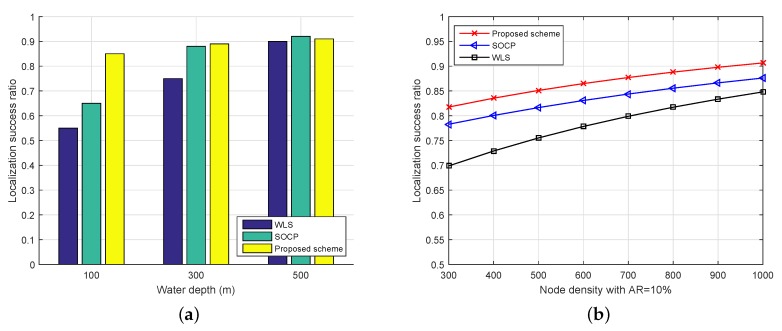
Comparison between the localization schemes. (**a**) Impact of water depth; (**b**) Influence of node density.

**Table 1 sensors-19-03105-t001:** Parameter settings.

Parameter	Value
Total number of nodes	$N_{node}=500$
Maximum communication radius	$R_{\mathrm{com}}$ = 100 m
Anchor ratio	AR = 3–12%
Geometrical spread coefficient	k=1.5
Multilateration type	Trilateration, M=3
Trilateration error	$\delta_{d}=36$ m
Maximum hop counts	$N_{h}=10$

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
