# Peer review of "An Efficient RSS Localization for Underwater Wireless Sensor Networks"

_sensors, 2019, doi:10.3390/s19143105_

Round 1

Reviewer 1 Report

Very good paper, stating clearly your proposed RSS-based event driven method for underwater localization. I had no particular difficulty in understanding the paper. However, It would be helpful if you summarize your findings. 

Reviewer 2 Report

In my opinion, this paper raises several concerns, which are related to the quality of the English, the clarity, and the technical content.

As far as the quality of the presentation is concerned, there are many typos, badly constructed sentences and misused words. Consequently, the reading is very difficult and unpleasant. I definitely suggest an extensive proofread by a qualified expert.

As for the technical content, the paper does not appear solid. For instance:

- page 8. Fig. 2 shows curves with three different symbols; however the legend shows only two of them. The difference among the three curves is not clear.

-  page 8. It is not clear why the distance error is assumed proportional to d.

-  page 8, line after (15). If I am not wrong, the parameter eta (which is assumed known by the Authors) is not defined.

- page 8. Eq. (16) depends on the parameter Beta’ (Beta prime) whereas the text reports the definition of Beta. In any case, the origin of (16) in not explained.

- page 9. The content of figure 3 is not clear. The text refers that “the vertical and the horizontal axes represent the total errors during the measurement step and the regression process, respectively”. If the vertical and the horizontal axes represent the error, where is the regression represented?

- page 9. The sentence “For a given network, a node is called Nh-hops localizable if it can be localized by using only measurement information at most Nh-hop references”  is unclear. The concept of Nh-hop localization was not explained before.

- page 10. The sentence “We also assume that the distance between normal sensors and the surface anchor nodes should meet the connectivity requirement”  is in contrast with what is reported in table 1. In fact, the communication range is 100 m, whereas the water depth is 600 m.

- eq (19). The argument of a cosine function MUST have no physical dimension, whereas the argument of (19) is expressed in seconds.

- Both in the text (line 235) and in figure 5 the attenuation is expressed in dBm, which is completely wrong. In the logarithmic notation, attenuations MUST be expressed in dB.

More in general, the paper does not appear completely sound and does not provide enough interesting material for a journal.

Reviewer 3 Report

In this paper authors have presented an event-driven RSS-based localization in UWSNs. A fast localization algorithm has been designed by exploiting a linear fitting model for RSS-distance relationship among the sensors, which is based on the centralized paradigm.

Overall, the design is good and timely. Simulation results show higher localization accuracy, which is a good feature of the design and a desirable one. Another novel contribution of the paper is the derivation of the path loss exponent.  Many case studies are also presented. Overall, I recommend acceptance of paper with minor revisions.

Round 2

Reviewer 2 Report

In my first review, I raised several concerns regarding the quality of the English, the clarity and the technical content.

Surprisingly, the Authors did not take (almost) any action to address my comments.

As for the quality of the English and the clarity, the paper has not improved. NO MODIFICATION has been made to the text in the first seven pages, which are still full of typos, badly constructed sentences and misused words. Indeed,  I did not ask to modify the paper organization, as written by the Authors in their reply. Instead, I suggested an extensive proofread in order to fix the many errors that make the reading really unpleasant and, in some cases, the content unclear. Even when the text has been modified, new errors have been introduced. For instance, at page 8 the Authors write: “In this figure, the solid yellow line interpolating ….”. Well, the figure does not show any yellow line.

As far as the technical content is concerned, in my first review I asked some clarifications. In most cases, the Authors did not address my requests.

For instance:

- page 8. I asked why the distance error is assumed proportional to d. They answered that this assumption “is clear for two reasons: (i) it focuses on the variables d^ and d; and (2) the values of d^ and d are nonzero.”. This answer makes no sense. In order for this assumption to be sound, the Authors should explain why it is reasonable: is it what happens in real underwater localization systems? If this is true, some references must be provided to support such claim.

- page 8. If I am not wrong, the parameter “Beta prime” that appears in (16) is not defined.

-page 9. As I wrote in my first review, the content of Figure 3 is unclear. As far as I can understand, this figure shows the actual distance, the estimated distance using raw data and the adjusted distance. This is, at least, what the Authors write in the figure legend. However, this is not consistent with the labels of the horizontal and vertical axes, which mention horizontal and vertical errors. “Distances” (as reported in the legend) and “distance errors” (as reported by the axes labels) are different things.

- with reference to (19), I insist that the argument of the cosine function must have no physical dimension, as it represent radians. This is a basic concept of trigonometry. This means that, at least, the coefficient 0.4 has a physical dimension, given by [1/s]. This should be reported in the paper.

- With reference to (2), please observe that P(d) is a power expressed in logarithmic units, hence it represents dBm or dBw (dB is definitely wrong). Moreover, the definition of term A(d,f) that appear in (12) is different from the definition given in [7]. Why?

As a general suggestion, it is good practice to reply to a reviewer addressing point by point his/her specific comments. Some of the responses provided by the Authors had nothing to do with the raised comments. Moreover, unless well-motivated circumstances exist, for each answer provide by the Authors, a modification of the text is required in the paper. In fact, the Reviewer’s doubt could be the same of a generic reader.

Author Response

Please find the attached file for detail

Round 3

Reviewer 2 Report

The Authors significantly improved the quality of the paper, fixing some errors and introducing further explanations.

The quality of the English has been improved as well, which makes the paper more readable. 

Minor grammatical issues remain, which need to be fixed before publication.

Author Response

Thank you very much for your detailed and helpful comments to improve the quality of the manuscripts.
In the revised manuscript, we made some minor modifications in order to improve the paper clarity.